# Generation of Orthotopic Patient-Derived Xenografts in Humanized Mice for Evaluation of Emerging Targeted Therapies and Immunotherapy Combinations for Melanoma

**DOI:** 10.3390/cancers15143695

**Published:** 2023-07-20

**Authors:** Chi Yan, Caroline A. Nebhan, Nabil Saleh, Rebecca Shattuck-Brandt, Sheau-Chiann Chen, Gregory D. Ayers, Vivian Weiss, Ann Richmond, Anna E. Vilgelm

**Affiliations:** 1Department of Pharmacology, Vanderbilt University, Nashville, TN 37232, USA; chi.yan@vanderbilt.edu (C.Y.); nabil.saleh@vanderbilt.edu (N.S.); rebecca.shattuck-brandt@vanderbilt.edu (R.S.-B.); 2Department of Veterans Affairs, Tennessee Valley Healthcare System, Nashville, TN 37232, USA; carolyn.nebhan@gmail.com; 3Division of Hematology & Oncology, Vanderbilt University Medical Center, Nashville, TN 37232, USA; 4Department of Biostatistics, Vanderbilt University Medical Center, Nashville, TN 37232, USA; sheau-chiann.chen@vumc.org (S.-C.C.); day.ayers@vumc.org (G.D.A.); 5Department of Pathology, Microbiology and Immunology, Vanderbilt University Medical Center, Nashville, TN 37232, USA; vivian.weiss@vumc.org; 6Department of Pathology, Ohio State University, Columbus, OH 43210, USA; 7Pelotonia Institute for Immuno-Oncology, The Ohio State University Comprehensive Cancer Center—Arthur G. James Cancer Hospital and Richard J. Solove Research Institute, Columbus, OH 43210, USA

**Keywords:** patient-derived xenograft, humanized mice, targeted therapy, immunotherapy, melanoma

## Abstract

**Simple Summary:**

Patient-derived xenografts (PDX) are valuable models in preclinical oncology drug development. However, they are not well suited for testing immune-based therapies. In contrast, when humanized mice are developed by xenotransplantation of irradiated mice with human CD34+ stem cells to allow for the growth of human tumor tissue in the context of a humanized immune system, it is possible to evaluate the response to both targeted therapy and immune therapy. Unfortunately, humanized mice transplanted with human PDX tissue frequently develop graft-versus-host disease (GVHD). Here, we optimized a protocol for generating humanized PDX mouse models with considerably reduced development of GVHD, making preclinical trials with immune checkpoint inhibitors possible.

**Abstract:**

Current methodologies for developing PDX in humanized mice in preclinical trials with immune-based therapies are limited by GVHD. Here, we compared two approaches for establishing PDX tumors in humanized mice: (1) PDX are first established in immune-deficient mice; or (2) PDX are initially established in humanized mice; then established PDX are transplanted to a larger cohort of humanized mice for preclinical trials. With the first approach, there was rapid wasting of PDX-bearing humanized mice with high levels of activated T cells in the circulation and organs, indicating immune-mediated toxicity. In contrast, with the second approach, toxicity was less of an issue and long-term human melanoma tumor growth and maintenance of human chimerism was achieved. Preclinical trials from the second approach revealed that rigosertib, but not anti-PD-1, increased CD8/CD4 T cell ratios in spleen and blood and inhibited PDX tumor growth. Resistance to anti-PD-1 was associated with PDX tumors established from tumors with limited CD8+ T cell content. Our findings suggest that it is essential to carefully manage immune editing by first establishing PDX tumors in humanized mice before expanding PDX tumors into a larger cohort of humanized mice to evaluate therapy response.

## 1. Introduction

Melanoma, the deadliest form of skin cancer, is a significant health concern. With approximately 100,000 new cases diagnosed annually, it represents the fifth and sixth most common cancer diagnoses in men and women, respectively [1,2]. Once melanoma has spread to lymph nodes or distant sites, the 5-year survival rate plummets to approximately 20% (2007–2013) [3]. The current standard of care therapies for metastatic melanoma includes immune checkpoint inhibitors (ICIs), a novel class of therapeutics that inhibit cytotoxic T lymphocyte antigen-4 (CTLA-4) or the programmed death-1 receptor/ligand (PD-1/PD-L1). Mechanistically, ICIs relieve the inhibitory signals to the immune system, allowing T cells to eliminate malignant cells. This host-directed mechanism is distinct from targeted therapies, which act on the level of the malignant cells. Currently, there are four FDA-approved immunotherapy drugs for use in patients with advanced or metastatic melanoma: ipilimumab, an inhibitor of CTLA-4; pembrolizumab and nivolumab, both inhibitors of PD-1; and atezolizumab, an inhibitor of PD-L1. These therapies have prolonged progression-free and overall survival in melanoma patients, with a median overall survival (OS) of 49% and a melanoma-specific survival of 56% at 6.5 years for patients treated with a combination of CTLA-4 and PD-1 inhibition [4]. Furthermore, with six FDA-approved checkpoint inhibitors clinically used in treating 18 different tumor types in nearly 80 different clinical settings, the application of ICIs in cancer medicine is rapidly expanding. Unfortunately, while some patients experience long-term durable responses at five years and beyond, a significant proportion of tumors progress on ICI [5,6]. 

Better approaches need to be evaluated; however, this is difficult using mouse cancer models because the preclinical evaluation of ICIs requires an intact immune system; therefore, models using human tumors established in immunodeficient mice are not applicable. Murine tumors established in immunocompetent mice are widely used for preclinical immuno-oncology research; however, they do not fully replicate the complexity and heterogeneity of human tumors, nor do they use the human immune system. Compared to the myeloid-cell-rich human immune system, B and T cells dominate the murine immune system [7]. To maximize translational utility, an optimal preclinical model should involve the growth of human tumor tissue in the presence of a fully competent human immune system capable of mounting anticancer immune responses for specific immunotherapeutic interventions. Humanized mouse models implanted with human tumor tissue are attractive for evaluating combinations of immune checkpoint blockade and targeted therapies to develop strategies to improve the survival of melanoma patients that do not respond to ICI alone. The protocols typically involve establishing PDX in NSG mice then transplanting the PDX into humanized mice established by engrafting human CD34+ hematopoietic stem cells into myeloablated immunocompromised mice. Therefore, time must be allocated for sufficient tumor growth and maintenance of engraftment to allow drug studies [8]. Several groups have reported outcomes with solid tumors (summarized in Table 1), but such studies are limited by cost, unstable engraftment, and the development of graft-versus-host disease (GVHD) [9,10,11,12]. GVHD occurs when the donor T cells attack recipient tissues and this is followed by antigen-presenting cell activation, expansion of alloreactive T cells, and a cytokine storm that results in tissue injury and destruction [13].

Our group has a long-standing interest in using humanized mice to develop strategies to improve the effectiveness of novel immunomodulatory anticancer agents [14]. The aim of the present study is to develop and characterize a PDX model in humanized mouse, with manageable toxicity, long-term human melanoma tumor growth, maintenance of human chimerism, that is suitable for testing the response to immune-based therapies. Here, we describe two different approaches for generating humanized mouse models for preclinical immuno-oncology research. In the first approach, tumor fragments were initially implanted into immune-deficient mice then engrafted into humanized mice that we established using cord blood CD34+ cells. Using this approach, we observed high engraftment efficiency and chimerism, but a GVHD-like phenotype developed consistently and rapidly. This was accompanied by rapid expansion of the human T cell population response to the tumor and early death of the engrafted humanized mouse. 

We postulated that when human tumor expands in an environment that has no antitumor immunity, then transferred to a mouse with a humanized immune system, there is increased risk for developing GVHD due to the lack of prior selection against human antigens or tumor neoantigens during the initial PDX outgrowth. Based on this reasoning, in our second approach, we sought to determine whether we could reduce or delay GVHD in our preclinical trials by first implanting fresh patient tumor fragments directly into humanized mice, letting immune editing occur as the PDX becomes established, then expanding the PDX into sufficient humanized mice for preclinical trials. Under these conditions, the humanized mice bearing PDX tumors were viable with little evidence of GVHD, had moderate engraftment efficiency and were suitable for preclinical studies to evaluate ICI-focused melanoma regimens in combination with novel targeted therapies. Given that GVHD is one of the major drawbacks for preclinical studies in humanized mice, this study describes improved methodologies for reducing GVHD for use of humanized PDX tumor models in preclinical studies to evaluate emerging immuno-oncology treatments in human settings.

## 2. Materials and Methods

**Patient tissue:** Tumor samples were collected on a tissue collection protocol approved by the Vanderbilt University, and all patient donors signed an approved informed consent before providing tissue samples. Informed consent to obtain tumor tissue was obtained from patients undergoing routine treatment at Vanderbilt University Medical Center (VUMC IRB #030220). Metastatic melanoma tumor tissue was obtained fresh, divided into ~0.5 × 0.5 mm aliquots, and stored in FBS + 10% DMSO in liquid nitrogen until implant. For this study four patient tumors were analyzed for melanoma associated mutations and three of the four were characterized and analyzed (Table 2): 1351, 3125, 3145 and 3101. 

Patient 1351 received the following treatments over a two-year period prior to surgery: imiquimod cream; anti-PD1; vemurafenib; steroids; temozolomide; trametinib; dabrafenib. At the time of progression on dabrafenib, a small sample of the removed tumor tissue was received for PDX implantation. For patient 3145, an aliquot of a metastatic lesion obtained from metastatic nodes was acquired for establishing a PDX prior to initiation of immunotherapy. Patient 3101 had a kidney transplant due to rhabdomyolysis. Eight years later, non-melanoma cutaneous malignancies of the head and neck were removed, and a dermal nodule of the right forearm was treated with cryotherapy. The patient presented two year later with bulky right axillary lymphadenopathy. We received tissue from the resection of the axillary lesion for establishing the PDX. This patient was immune suppressed. Slides of FFPE tumor tissue from all the patients were stained for CD3 and CD40 to access the likelihood of the response to anti-PD1 and rigosertib (RGS) therapy, based upon prior observations that induction of CD40 is part of the mechanism of the response to RGS [15] and it is well established that a strong infiltration of CD3+ T cells is associated with the response to anti-PD1 [16].

**Mouse Studies:** Animal studies were approved by the Vanderbilt University Institutional Animal Care and Use Committee (IACUC; M2000008-00, V1700194-01). NSG mice (NOD.Cg-*Prkdc^scid^* Il2rg^tm1Wjl^/SzJ) were purchased from Jackson Labs, and NSG-SGM3 (NSGS, NOD.Cg-*Prkdc^scid^ Il2rg^tm1Wjl^* Tg (*CMV-IL3,CSF2,KITLG)1Eav/MloySzJ*) mice were also purchased from Jackson Labs. HuNOG-EXL mice were donated or purchased from Taconic. Mice were maintained in sterile housing within a level 1B barrier facility. HEPA-filtered sterile caging was used where the interior cage height was 5.5 inches, and the interior floor area was 75 square inches. Sterile food and water were provided. Mice were given diet gel treats if they exhibited a loss of weight. Cardboard huts were used for enrichment. 

For humanization of NSGS mice, 3-4-week-old mice were placed in an RS2000 X-ray biological irradiator without restraint and irradiated with 179 cGy/min to obtain a whole-body level of 1–2 Gy in the level 1 barrier facility and 24 h later mice were implanted with 1 × 10^5^ CD34+ (Lonzo) cells from human cord blood via retroorbital injection. Mice underwent blood sample collection from the tail vein to confirm presence of huCD45+ cells that develop from differentiation and expansion of the huCD34+ cells. The first blood sample was taken 3 weeks after injection of CD34+ cells and subsequent samples were taken approximately 9 weeks, 15 weeks, and 21 weeks after injection. If CD34+ engraftment was unusually delayed, the sampling is extended after 21 weeks. Mice that had a minimum of 25% huCD45+ cells in circulation based upon flow cytometric analysis received human tumors. Surgical implantation of tumor tissue was performed using a sterile technique as described previously [17]. Briefly, tumor was minced in a Petri dish to ~1 mm fragments, which were implanted at bilateral flanks of anesthetized mice (isofluorane). The surgical site was closed with a staple, which was removed 10 days post-operatively. Mouse body weight was measured once per week until the development of palpable tumors, at which point both body weight and tumors were measured twice per week. Tumor dimensions were measured with microcalipers. Tumor volume was estimated as 0.5 × length × width × width.

The huNOG-EXL mice were purchased from Taconic at 18 weeks post-engraftment with CD34+ cord blood cells. Taconic routinely engrafts 100,000 huCD34+ cells into irradiated mice. They have determined that engraftment rates of ≥25% are generally saturated with 1 × 10^5^ cells in huNOG-EXL mice. Only mice with >25% engraftment are sold and shipped. Chimerism was confirmed approximately 3 days after the mice were received into the Vanderbilt level 1 barrier facility. Those mice with confirmed greater than 25% human chimerism were implanted with melanoma tumors using the same protocol as for the huNSGS mice. Human tumor fragments from patients 3101, 3145 and 3125 were each implanted into nine huNOG-SGM mice that had been engrafted with huHLA-A2+ CD34+ cells (3 mice for each tumor). Tumors only grew out in the 3101 and 3145 patient implants and only patient 3101 tumor reached sufficient size to transplant into 20 additional huNOG-EXL mice. The mice were 22 weeks post-engraftment with CD34+ cord blood cells (HLA-A2-) with a verified chimerism of >25%. Chimerism was continuously followed over the course of the experiment and when the tumors reached a volume of 50–100 mm^3^, treatment was initiated with vehicle + IgG control (Bio X Cell, #BD0089), vehicle + anti-PD1 (BioXCell # BE0146, 200 μg twice a week), RGS (Onconova, 300 mg/kg QD 5 days/week), or RGS + anti-PD1 (300 mg/kg QD and 200 μg twice a week, respectively). Treatments continued until tumors reached 15 mm in diameter or mice required humane sacrifice due to other humane endpoint (i.e., weight loss >20% initial body weight, poor body condition score, or decreased activity). With this methodology we were able to analyze therapeutic responses in humanized mice implanted with human melanoma tumors for an average of 49 days. At the end of the experiment, the final weight of the tumors was recorded, and tumors were either flash-frozen or processed for staining and subsequent flow cytometry.

GVHD was assessed as follows: Since GVHD is expected to present signs as early as 30 days post-procedure we monitored mice weekly for the first 25 days post-implantation, then starting on day 25, animals were monitored 2 times per week. Pre-procedure weights were collected and used as baseline weights. The mice were assigned a grade on the GVHD assessment scale based on an assessment of five conditions: weight loss, posture, activity, skin integrity, and fur texture. Mice received a score of 1 for each positive symptom, with a mild symptom receiving a point of 1 and a more severe symptom a point of 2. Humanized mice with a clinical score <3 were defined as subclinical GVHD, and mice with a clinical score 3 were defined as clinical GVHD. Mice with GVHD scores between 3 and 6 were provided DietGel in the home cage, and if needed for dehydration or skin issues, a veterinarian was contacted. If a mouse had a GVHD score >6, the veterinarian was contacted and, if necessary, the animal was euthanized. If additional signs or symptoms of distress were observed (changes in respiration, self-mutilation, guarding, abnormal ambulation), we weighed the animal, and if the animal lost >20% from baseline, the lab contacted a DAC veterinarian or euthanized the mouse.

**Flow cytometry:** Flow cytometric analysis of human and mouse immune cells was performed as described previously [17,18,19]. Briefly, white blood cells were collected by lysing red blood cells using ACK lysing buffer. To obtain single-cell suspensions, spleens were pressed through a cell strainer and washed with PBS, and lung, livers and tumors were enzymatically digested and mechanically dissociated using gentleMACS dissociator (Miltenyi) according to the manufacturer’s recommendation. Bone marrow cells were flushed from femurs using a syringe filled with PBS. To obtain cells from the peritoneal cavity, euthanized mice were injected with 10 mL of PBS into the peritoneal cavity and gently massaged, followed by the aspiration of PBS containing dislodged cells. Viability staining was performed using LIVE/DEAD™ Fixable Aqua Dead Cell Stain Kit or eBioscience™ Fixable Viability Dye eFluor™ 780 (Thermofisher) or Zombie NIR™ Fixable Viability Kit (Cytek) according to the manufacturer’s protocol. Cells were incubated with fluorescently labeled antibodies indicated in Appendix A for 30 min, washed twice, and analyzed on BD LSRFortessa™ or Cytek Aurora Cell Analyzer. Antibodies were purchased from BioLegend. The resulting FCS files were uploaded to OMIQ cytometry analysis platform for UMAP-map construction and FlowJo for traditional gating and t-SNE analysis.

For chimerism analysis, approximately 100 μL of peripheral blood was obtained by either submandibular puncture using lancet or by tail vein nick using microvette capillary tubes (Sarstedt Inc #16.444.100). As a positive control, an equivalent volume of blood was obtained from wild-type C57Bl/6 mouse (mouse control) or healthy human donor (human control). After pelleting sample, erythrocytes were lysed by incubation in an ammonium-chloride-potassium buffer for 5 min at room temperature. Cells were then washed and stained with viability dye (1:1000 in PBS) for 30 min at room temperature, then blocked in Fc blocker (BD #553142) in FACS staining buffer (PBS supplemented with 2.5% FBS and 0.5% EDTA). After blocking, cells were stained with PerCP/Cyanine5.5 anti-mouse CD45 Antibody (1:1000; Biolegend Cat# 103132) and APC anti-human CD45 Antibody (1:200; Biolegend Cat# 368512) for 20 min at 4 °C protected from light. After incubation, cells were washed twice in FACS staining buffer and fixed in 300 µL 1% formalin in PBS. Data were collected using a BD LSR Fortessa flow cytometer and analyzed using FlowJo software (Version 10.5.3).

**Drug treatment**: RGS (300 mg/kg, Onconova) was reconstituted in water and administered five days per week by oral gavage in a total volume of 100 uL. Checkpoint inhibitor antibody and isotype control were purchased from BioXcell. Anti-human PD-1 (clone: RMP1–14) or an equivalent amount of isotype control, Rat IgG2a control (clone: 2A3), were administered intraperitoneally at 200 μg per mouse twice per week for the duration of the experiment.

**Immunohistochemistry staining:** Hematoxylin and eosin (H&E), human CD3, human CD40 and human CD8 immunohistochemical staining of 10% buffered formalin-fixed, paraffin-embedded tissue sections were performed by the VUMC Translational Pathology Shared Resource (TPSR) as described previously [15,20]. Tumor tissues were stained with anti-human CD3 (Abcam ab16669 clone SP7) and anti-human CD40 (Abcam ab13545) antibodies. Results from human tonsil and mouse spleen were included as positive and negative controls, respectively. Immunohistochemical staining was performed on lung tissue using a combined anti-human Melan A/Mart1 rabbit monoclonal antibody red stain (clone MLANA/1761R; NSJ Bioreagents Cat# V7268IHC) to identify melanoma metastases. H&E and immunohistochemical stains were reviewed by a board-certified pathologist (VW). Evaluation criteria for melanoma metastases included cytologic and architectural atypia consistent with melanoma and positive staining for the combined anti-human Melan A/Mart1 rabbit monoclonal antibody red stain.

**HLA typing**: HLA typing was performed by Murdoch University Institute for Immunology and Infectious Diseases.

**Statistical analysis:** Power analyses were performed which revealed that an *n* = 5 per group measured over time has sufficient power to detect meaningful differences among groups where such differences are present. Data were summarized in figures using the mean ± S.D. for the error bar chart. The mean of the two groups was compared using Welch’s *t*-test test. For tumor volume over time, data were analyzed on a natural log scale. A linear mixed-effect model was used to account for the correlation structure among repeated measures from the same mouse. Using model-based (least-square) means, the average tumor growth rate for each treatment was estimated and compared using the Wald test. The Holm correction was used to adjust the *p* value for pairwise comparisons. The reports of statistical significance indicated * for *p* < 0.05, ** for *p* < 0.01, and *** for *p* < 0.001, respectively. The analyses were performed using R version 4.1.2 or GraphPad Prism software.

## 3. Results

### 3.1. Characterization of the CD3 Infiltrate within the Patient Melanoma Tumors Used for PDX Engraftment

The distribution of CD3+ T cells and CD40 expression was evaluated from nine melanoma patients with stage III disease and one stage II disease. From these data, two melanoma tumors were selected for study in the humanized model based on CD40 and CD3 staining. Tumor from patient 3145 was selected based on areas of high CD3 T cell infiltration in the peri-tumoral areas with some infiltration into the tumor proper. CD40 staining was variable with some areas of high intensity (+3) and other areas of moderate staining (+2). CD40 staining was both cytoplasmic, nuclear and at the plasma membrane of the tumor cells. Patient 3101 tumor was mostly devoid of CD3+ T cell staining, though there were areas with a few clusters of T cells in blood vessels and areas of the tumor where tumor cells appear to have died. CD40 staining for patient tumor 3101 was moderate (+2) with occasional areas of intense staining around blood vessels (+3). The CD40 staining was nuclear and cytoplasmic, but not readily apparent at the plasma membrane (Figure 1). HLA typing of these two tumors revealed that both carry the HLA-A2 allele (3145: 23:01:01G; 3101: 32:01:01G).

### 3.2. Characterization of huNSGS Mice Generated Using Cord-Blood-Derived CD34+ Hematopoietic Stem Cells

To generate the humanized mice for preclinical immuno-oncology studies, we used triple transgenic NSGS mice (available from Jackson Labs) expressing human IL3, GM-CSF, and SCF to support human hematopoietic cell engraftment. Mice were treated with 100 Rad whole-body gamma irradiation at four weeks of age, then transplanted with 1 × 10^5^ human cord blood-derived CD34+ hematopoietic stem cells (purchased from Lonza) via retroorbital injection (Figure 2A). We tested stem cells from two distinct donors. Ten weeks after injection, engraftment was measured using flow cytometry of bone marrow suspensions detecting mouse and human CD45+ cells (Figure 2B). We also measured human chimerism in various organs, including blood, spleen, peritoneal cavity, bone marrow, lung, and liver (Figure 2C). The highest levels of human cell engraftment were observed in bone marrow (75–93% of human CD45+ cells), spleen (81–88%), lung (83–91%), and liver (86–91%). Intermediate human chimerism was detected in the peripheral blood (30–70%). Of all tested tissue sources, peritoneal cavity lavage samples had the lowest level of human immune cells (1–27%) (Figure 2C). With this protocol, mouse irradiation followed by engraftment of naïve CD34+ stem cells from cord blood, results in a mouse that is somewhat immune naïve. It is thought that transplant of a human tumor with a different HLA type into an immunologically naïve mouse can minimize GVHD or increase time to onset of GVHD.

### 3.3. Establishment of Melanoma PDXs in huNSGS Mice (huNSGS-PDX)

In the second set of experiments, four NSGS mice were engrafted with human CD34+ immune cells from donor 3, distinct from donors 1 and 2 described above, to develop huNSGS mice. After allowing a sufficient time for engraftment to occur, mice were implanted with PDX-1351, previously established by our group [17,21,22]. Patient 1351 was positive for the HLA-A2 allele (03:01:01G). This PDX was established and passaged once in non-humanized NSGS mice to expand the PDX tissue. Tissue fragments of the passaged PDX were implanted into huNSGS mice to generate huNSGS-PDX mice (Figure 2D).

Based on our prior experience, non-humanized NSG mice implanted with PDX-1351 tumors recovered well after surgery and showed no signs of health problems or toxicities for the duration of drug treatment experiments [21]. In contrast, huNSGS-PDX mice implanted with PDX-1351 tumors developed a toxicity phenotype characterized by severe weight loss, alopecia, and lethargy by two weeks post-implantation; however, a comprehensive histological evaluation was not performed at that time. Ultimately, all huNSGS-PDX mice died or required humane sacrifice due to the extensive weight loss and poor health conditions 1–4 weeks post-engraftment. Blood and organs from mice were subjected to chimerism analysis as described above using flow cytometry with mouse- and human-specific anti-CD45 antibodies. As seen previously, mice exhibited moderate human chimerism in the blood, while high levels of human immune cells were detected in tissues, such as the spleen and lung (Figure 2E). Of note, tumors from these mice were predominantly infiltrated with mouse CD45 cells, while human CD45 cells constituted 5–40% of the total CD45+ immune cell population (Figure 2F). Moreover, human CD45+ cells constituted only approximately 1–3% of total viable cells in tumor cell suspensions (Figure 2G). These data suggest that human immune cells preferentially accumulated and potentially reacted to mouse tissues with much less homing to target human tumors. These factors may have contributed to the severe weight loss, alopecia, and wasting phenotype in these mice.

### 3.4. Identification of Human Immune Cell Subsets in huNSGS Mice

In addition to antibodies recognizing human and mouse CD45+ cells, our flow cytometric analysis included lineage markers of key immune cell subsets, as indicated in Appendix A. Using the multicolor cytometry assessment of surface marker expression coupled with the UMAP dimension reduction analytical approach, we identified and quantified subsets of T, B, NK cells, and myeloid cells in the blood and various tissues collected from humanized mice that did not receive tumor engraftment as shown in Figure 3A,B. Blood samples from wild-type C57Bl/6 mouse (Figure 3A, left panel) and a human donor were assayed as positive controls (Figure 3A, right panel). We manually gated human B, T, NK, and CD11b+ myeloid cells as shown in Appendix A and compared their relative abundance in huNSGS and huNSGS-PDX mice reconstituted with hematopoietic stem cells from three distinct donors described above (donors 1-2–huNSGS, shown in Figure 2C, donor 3–huNSGS-PDX shown in Figure 2E,F). We found that human CD3+ T cells were abundant in the blood of humanized mice established from donors 2 and 3, but not donor 1. In contrast, human CD19+ B cells were enriched in spleens of mice engrafted with all three donors. High engraftment of human CD11b+ myeloid cells was observed in the lungs of huNSGS and huNSGS-PDX mice (Figure 3C). Human NK cell percentages were low across all tested tissues. We also noted that bone marrow composition varied greatly in a donor-dependent manner and there was also a notable mouse-to-mouse variability. It is puzzling that the same donor would have different patterns of subpopulations of cells in different humanized mice. Clearly, this is one of the issues in evaluating the quality of the humanization procedure and data should be evaluated in reference to these variabilities. For example, mouse 3 from donor 3 in Figure 3C shows considerably fewer CD3+ T cells in the blood, as compared to mouse 1 and 2. There are comparable CD3+ cells in the spleen for mouse 1 and 4, yet we did not detect CD3+ cells in the bone marrow of this mouse and CD3+ T cells were barely detectable in the lung. The percentage of B cells in the spleen was fairly constant but varied in the blood. Moreover, tumor variability in B cells, T cells and myeloid cells was also quite variable.

There are two possibilities for these differences: (1) the CD34 cells may vary in the direction of differentiation from mouse to mouse as well as in the infiltration of specific organs; (2) there may be variability from mouse to mouse in the recovery of the T cells across tissues for the flow cytometric analysis protocol. That said, when patient tumors are analyzed from different metastatic sites, there is often considerable tumor heterogeneity and variability in the makeup of the CD45+ cells in the tumor [23,24,25,26]. Similarly, tumors from huNSGS-PDX mice displayed high degree of variability of human leukocytes distribution (Figure 3C, lower right panel). Human leukocytes in the tumor are most likely originated from the engrafted donor CD34+ cell, as several groups have demonstrated the loss of tumor-infiltrating immune cells in the process of PDX expansion in non-humanized mice [27,28,29]. Overall, tumors contained very few human immune cells (Appendix A), suggesting minimal immune cell recruitment into the tumor. Intriguingly, all the mice that received stem cells from donor 1 lacked T cells. Lack of T cells was associated with a relative increase in the B cells in all tested tissues (Figure 3C). It is plausible that the lack of T cells in these mice was a result of the unique properties of stem cells from this stem cell donor. In contrast, mice implanted with stem cells from donors 2 and 3 developed all key immune subsets, including T cells (Figure 3C). These findings suggest that the success of human immune system reconstitution in humanized mice is donor dependent as related to the qualities of hematopoietic stem cells used for humanization. As such, the quality of engraftment should first be assessed in blood, spleen, bone marrow, liver, lung, and peritoneal cavity before using that donor for preclinical trials huPDX trials.

We studied T cells in humanized mice engrafted with stem cells from donors 2 (*n* = 1) and 3 (*n* = 4) in greater detail. As described above, mice reconstituted with stem cells from donor 3 were implanted with melanoma PDX tumor. Only mice that received vehicle control treatment (no drugs) were included in this analysis. Human PBMC, as well as PBMC and splenocytes from immunocompetent C57Bl/6 mice were used as normal controls to identify immunome changes characteristic of humanized mice. The gating strategy used to identify lymphoid immune cell subsets in wild-type mice is shown in Appendix A. We observed that CD4+/CD8+ T cell ratio was skewed in favor of CD4+ T cells in the blood and organs of humanized mice (Figure 3D). Others have previously reported increased CD4+ T cells as compared to CD8+ T cells after engraftment [30]. Future follow-up studies are needed to explain the CD4+ pre-dominance observed here. Another observation was that a high percent of T cells infiltrating tissues and organs of humanized mice expressed activation markers, such as CD69 and HLA-DR (Figure 3E,F). These findings may indicate that T cells reacted toward the host cells, which may explain the toxicity phenotype observed in huNSGS mice.

### 3.5. Analysis of Human Myeloid Cell Populations in Humanized Mice

Another notable observation from the immunophenotyping analysis was that the composition of immune subsets in humanized mice was distinct from that in human and wild-type immunocompetent mice. Specifically, the relative percentages of CD11b+ myeloid cells were reduced in the blood of humanized mice compared to human blood or wild-type mouse blood (13–25% in blood from humanized mice vs. approximately 60% in human and mouse blood) (Figure 4A, left panel). Bone marrow is a key site of myeloid cell hematopoiesis. Notably, the relative percentage of myeloid cells in the bone marrow of humanized mice was greatly reduced compared to wild-type C57Bl/6 mice (Figure 4A, right panel). However, the analysis of non-immune tissues revealed that myeloid cell populations were present in peritoneal cavities, lungs, and livers of humanized mice (Figure 3C). Based on these findings, two non-mutually exclusive scenarios are possible. The first is that our humanized mouse protocol favors the engraftment of lymphoid cells over myeloid cells. The second potential explanation is that human myeloid cells developing in humanized mice tend to immigrate from the circulation and accumulate in the distal organs.

Multicolor flow cytometric analysis focused on myeloid cell-specific markers showed that the total populations of CD11b+ cells in blood and organs of huNSGS and huNSGS-PDX mice included CD66b+ granulocytes and a subset of cells positive for MHCII (HLA-DR), FC receptors (CD64), and CD123, and CD11c markers known to be expressed by dendritic cells and macrophages (Figure 4B). Across all tested tissues, the highest levels of myeloid cell engraftment were observed in lungs and livers of humanized mice (Figure 4C). After gating individual myeloid cells subsets using strategy shown in Appendix A, we compared the relative abundances of granulocytes, monocytes, dendritic cells, and macrophages in the blood and organs of individual huNSGS mice implanted with stem cells from 3 distinct donors (Figure 4D). We also included a population of “other” CD11b+ cells that did not fit into the four categories above. Unlike lymphoid compartment where high degree of donor-dependent variability was observed in regards of the efficiency of T and B cell engraftment, myeloid subsets displayed relatively similar distribution across mice engrafted with 3 distinct donor stem cells. Bone marrow samples from huNSGS mice contained all tested cell subsets. Peritoneal cavity lavage samples had very few CD14+ monocytic cells, while CD64+ macrophages were seldom in blood and spleens (Figure 4D). Tumors from huNSGS-PDX mice displayed all myeloid subsets, without a strong pre-dominance of a particular cell type (Figure 4E). However, tumors had overall very few human immune cells; therefore, the data on human immune cell composition in tumors should be interpreted with caution.

Interestingly, we noted that the levels of circulating granulocytes were relatively lower in huNSGS mice compared to that in human blood. To understand whether there may be a defect in myelopoiesis present in humanized mice that limited granulocyte differentiation, we compared the composition of the bone marrow myeloid population in huNSGS mice to that in wild-type immunocompetent Balb/C animals. The gating strategy for identification of myeloid cell subsets in wild-type mice is shown on Appendix A. Surprisingly, the composition was similar in both models (Figure 4F). These findings suggest that a relatively low content of circulating granulocytes in huNSGS mice may be due to their immigration into tissues.

### 3.6. Xenograft Implantation into huNOG-EXL Mice

In several of our huNSGS mice we observed massive toxicity and GVHD once tumor was implanted. We surmised that the allogenic response against the melanoma tumors might indeed facilitate an anti-mouse xenogeneic reaction. Clearly, we had considerable infiltration of huCD3+ T cells in the lung and bone marrow with donor 3 as is shown in Figure 3C. This led us to examine whether if the tumor implants were first initiated in humanized mice, we might select against some of this reactivity since immune editing might resolve HLA differences in the initial outgrowth. One issue with using huNSGS PDX tumor bearing mice for preclinical studies or co-clinical trials is the expected lifespan of engrafted mice is only approximately 4 months due to fatal disease resembling hemophagocytic lymphohistiocytosis [31]. However, huNOG-EXL mice have stable engraftment for the lifetime of the mouse, making these mice potentially more suitable for our therapeutic studies with targeted therapies and ICI [32]. Moreover, since the GM-CSF and IL-3 are expressed at more physiological levels, there is less danger of macrophage activation syndrome. Therefore, as an alternative approach, we utilized the humanized NOG-EXL mouse model, hereafter huNOG-EXL, that is genetically engineered to express human GM-CSF and IL-3 cytokines to support myeloid lineage engraftment and thus better recapitulate the human immune system.

To evaluate the utility of huNOG-EXL model for ICI studies, we performed a pilot study by implanting three different human melanoma tumors that had never been implanted into mice (p0) into huNOG-EXL mice (HLA-A2+) to generate huNOG-EXL-PDX mice (Table 2). As a comparator, p0 tissue was implanted at the same time into non-humanized NSG mice (NSG-PDX). Two of the three p0 tumors (PDX-3101 and PDX-3145, both HLA-A2+) grew in huNOG-EXL mice. PDX-3101 grew in two of three huNOG-EXL mice, with a mean tumor take time (time to the presence of palpable tumor) of 95 days, compared to growth in NSG mice, with a mean tumor take time of 68 days (Figure 5A). Among the huNOG-EXL-PDX mice, two animals gained weight after tumor implantation (4.2% and 6.1%), while one animal lost weight consistently from day 78 when the tumor entered a phase of rapid growth until humane sacrifice at day 110 (due to >20% weight loss). Non-humanized NSG-PDX mice showed weight gain throughout the course of the experiment. Interestingly, tumor bearing mouse that had to be euthanized at day 110 developed a wasting phenotype characterized by alopecia and skin sloughing (Figure 6A, see the Toxicity section below). PDX-3145 grew in two of three implanted huNOG-EXL mice within a mean take time of 109 days, but only very small tumors developed, compared to growth in two of two NSG mice, with a mean tumor take time of 78 days (Figure 5B). However, at the time of experiment completion (day 133), all huNOG-EXL-PDX mice were trending towards weight loss (+0.5%, −6.7%, and −15.0% weight change) while non-humanized NSG-PDX mice showed stable or increased weight. It is important to note that in this experiment with PDX-3145 implanted tumor, the weight loss occurred independent of tumor growth in the huNOG-EXL-PDX mice, in contrast to the observations with the PDX-3101 tumor. At 100 days after tumor implant, huNOG-EXL-PDX chimerism was assayed from peripheral blood via submandibular puncture to evaluate the persistence of human immune cells in this model. In all mice, >25% of human CD45+ cells were observed within the total PBMC population, confirming persistent humanization (range, 35.7–82.4% human CD45+). Of note, the chimerism for humanized mice implanted with PDX-3101 was high, chimerism was much higher in mice bearing PDX-3145. While one might speculate that the increase in chimerism had a role in the reduced tumor growth for PDX-3145, this is not likely since the tumor also remained equally small in one of the NSG mice implanted with PDX-3145.

### 3.7. Toxicity

In this pilot study, one of the huNOG-EXL-PDX mice successfully grew p0 tumor tissue 126 days post-implant (PDX-3101) but also experienced a wasting phenotype characterized by weight loss, alopecia, and skin sloughing focused around cranial and caudal areas. This toxicity ultimately required humane sacrifice and necropsy on day 133 post-implant (Figure 6A). Necropsy demonstrated macrophage and multinucleated giant cell infiltrates in the liver of this animal, showing erythrophagocytosis and intracytoplasmic pigment. This pigment was confirmed to be iron hemosiderin. Microscopic evaluation of gross alopecia observed showed infiltration and effacement of hair follicles and hair bulbs by lymphocytes, macrophages and multinucleated giant cells **(**Figure 6B–E).

### 3.8. ICI and RGS Treatment of huNOG-EXL-PDX Mice

After demonstrating that when PDX-3101 tumor is initiated in huNOG-EXL mice these mice survive over 100 days and exhibit reduced incidence of GVHD-like disease, we next performed an expanded experiment to evaluate the activity of novel Ras-pathway inhibitor, RGS, combined with anti-PD-1 in this model. We have previously shown that RGS induces tumor cell expression of CD40 and enhances the response to anti-PD1 in several immune-competent mouse models of melanoma [15]. Since this experiment required a large number of humanized mice engrafted with the same donor, for convenience we purchased twenty huNOG-EXL mice and surgically implanted with p1 PDX-3101 tissue to generate huNOG-EXL-PDX mice as described in the Methods section.

Upon development of tumors ≥50 mm^3^, mice were randomized to the treatment groups. The median time to development of a sufficient tumor was 69 days (range, 59–115). Notably, 15% (3/20) of mice died prior to enrollment into therapy. One of the three mice spontaneously expired 10 days post-operatively with weight loss of 14.5% (24), and the other two mice experienced weight loss requiring humane sacrifice prior to the development of tumors ≥50 mm^3^ (5, 16). In total, 17/20 (85%) mice demonstrated sufficient tumor growth and health to undergo therapeutic treatment. These data indicate that, in contrast to experiments with PDX tumors that were initially established in a non-humanized NSG mouse prior to implantation into humanized NSGS mice where nearly 100% of mice exhibited wasting. Only 15% of the mice with PDX implants that were established in huNOG-EXL mice exhibited sufficient wasting to limit their enrollment in this treatment study.

Mice were iteratively enrolled to drug treatment with one of four treatment paradigms: (1) vehicle + IgG control, (2) RGS + IgG control, (3) vehicle + anti-PD-1 or 4) RGS + anti-PD-1. Mice (*n* = 5 per group) were treated 5 days per week with 300 mg/kg RGS by oral gavage (or equivalent volume H_2_O control) and twice per week with 200 μg anti-PD-1 or IgG control (see Methods for further information). The median time on treatment was 49.5 days (range, 20–96). Reasons for termination included weight loss (58.8%, 10/17), excess tumor volume (11.8%, 2/17), other humane endpoint (5.9%, 1/17), or spontaneous expiration (5.9%, 1/17). Three mice (17.6%, 3/17) were maintained until the termination of the experiment in good health with acceptable tumor volume. The tumors in the RGS + IgG showed a statistical difference in tumor growth on a natural log scale (*p* = 0.002) when compared to the vehicle + IgG control group showed a significant difference in growth, but PDX tumor 3101 was resistant to anti-PD-1 treatment and curiously, anti-PD-1 treatment eliminated the growth inhibitory response to RGS (Figure 7A). Of note, PDX tumor 3101 had a low level of CD3+ staining and high staining for CD40 (Figure 1) and did not exhibit mutations in BRAF or NRAS, but there were mutations in NF1, FGFR, and CDKN2A, (see Table 2 for mutation profile for all PDX tumors studied). Chimerism check pre- and post-treatment revealed >25% human CD45+ cells in peripheral blood in all mice pretreatment (100%, 17/17) with sufficient persistence in a majority of mice throughout treatment duration (88%, 14/16) (Figure 7B,C).

Flow cytometric analyses were performed to characterize immune cell subpopulations in the spleen (Figure 7D), blood (Appendix A), and tumor (Figure 7E) tissues. Among the isolated splenocytes, RGS and anti-PD-1 alone or in combination induced a significant 4~6-fold increase in the frequency of CD8+ T cells (Figure 7D). There were no significant changes in myeloid cell, dendritic cell, B cell, NK cell, total T cell, regulatory T cell or CD4+ T cell content of spleens in the response to treatments. Thus, the CD8/CD4 cell ratio increased significantly in the spleen of huNOG-EXL-PDX mice bearing PDX 3101 with RGS plus anti-PD-1 treatment (Figure 7D). Consistently, there was a trend (*p* = 0.0571) toward increased CD8/CD4 cell ratio in the blood of huNOG-EXL-PDX mice treated with RGS + IgG compared to the vehicle + IgG treated mice (Appendix A). Furthermore, either anti-PD-1 alone or in combination with RGS was able to significantly increase CD8/CD4 cell ratio in the blood. These results suggest that while the combination of RGS and anti-PD-1 did not result in an effective tumor growth inhibition of PDX-3101 established from an immune cold tumor in huNOG-EXL-PDX mice, these treatments were able to promote CD8+ T cell responses in the distant lymphoid organ and blood circulation, which is consistent with our previous findings in the syngeneic mouse models of melanoma [15]. Detailed t-distributed stochastic neighbor embedding (t-SNE) analysis of the tumor microenvironment (TME) flow cytometry data showed no significant alterations in the frequency of myeloid cell, B cell, NK cell and T cell content of tumors in the response to treatments (Figure 7E). Notably, the absolute cell count of CD8+ T cell per tumor tissue was limited to 1 in 100,000 of total live cells isolated from tumor tissues. Indeed, subsequent immunohistochemistry (IHC) staining of human CD8 antigen confirmed the limited CD8 cell content in the tumor of huNOG-EXL-PDX mice bearing PDX-3101 (Appendix A). We observe minimal to no cross-reactivity of the human CD8 IHC antibody in the mouse spleen samples of tumor-free mice and were able to identify positive CD8 signals in the human tonsil as positive control samples as well as the lung tissues of huNOG-EXL-PDX mice bearing PDX-3101 (Appendix A). As shown in Appendix A, the combinatory treatment of RGS and anti-PD-1 induced a significant increase (~10%) of the frequency of CD69+ activated CD8+ T cells compared to the vehicle + IgG treated mice in the blood circulation. The activation status of CD8+ T cells in the spleen of huNOG-EXL-PDX mice bearing PDX-3101 was not altered by treatment of either RGS and anti-PD-1 alone or in combination. Given the low CD8+ T cell content in the tumor, we did not obtain enough cells to analyze the activation status. Our study developed improved methodologies that enable the use of humanized PDX tumor models to predict patient response to targeted therapy, ICI, or the combination to evaluate emerging immuno-oncology treatments in human settings. In addition to the evaluation of primary tumor growth, our optimized humanized PDX tumor model may further provide therapeutic evaluation on lung metastases of melanoma tumors. Indeed, histologic evaluation of the lungs of huNOG-EXL mice bearing PDX-3101 tumors demonstrated normal-appearing lung parenchyma. Immunohistochemical staining using a combined anti-human Melan A/Mart1 red stain for human melanoma cells demonstrated a small, 1 mm, deposit of metastatic melanoma in the lung of a vehicle-treated mouse (Appendix A). Altogether, our data reveal that PDX-3101 responds to treatment with RGS but has an “immunologically cold” TME, lacking sufficient CD8+ T cells to respond to anti-PD-1, which likely contributed to the resistance of this PDX tumor to anti-PD-1 therapies.

## 4. Discussion

Mice transplanted with human immune cells are an attractive model system for the preclinical studies of anticancer therapies dependent on the immune system. A variety of sources of hematopoietic stem cells have been used for developing humanized mice, including CD34+ cells expanded from umbilical cord blood or bone marrow, CD34+ cells expanded from the peripheral blood of the patient, or T cells isolated from the peripheral blood of the patient whose tumor is engrafted into mice bearing the PDX. Ideally, CD34+ cells autologous to the PDX would be engrafted. However, this is not always possible due to the limited availability of clinical samples, so alternative measures must be used. In addition, engraftment with PBMCs or CD34+ cells expanded from PBMCs has the disadvantage of a short life before the onset of GVHD [33] and the repertoire of human immune cells in the mouse is limited (over-representation of CD4+ T cells and few myeloid-derived cells) [14]. High levels of CD69+ T cells have been observed in transplant patients who developed GVHD [34]. Studies show that engraftment of CD34+ cells from cord blood or bone marrow results in extended strong chimerism and a fuller representation of the human immune cell compartment. While some humanized models utilize implantation of human fetal liver CD34+ and thymus into the subrenal capsule, this methodology cannot be used in states that have banned human fetal tissue for experimentation.

Tumor selection can also influence the outcome. Tumors that grow as a PDX in mice are generally more aggressive; similarly, those patients tend to have poorer outcomes [35]. While engraftment of melanoma tissues into NSG mice has a >75% success rate, engraftment into huNSGS does not appear to occur at the same level. Moreover, GVHD development and tumor growth impairment based upon lack of proper HLA match remains a problem. Lifespan of NSGS mice may vary by donor and can be impacted by environmental and experimental factors. The onset of GVHD is highly donor-dependent and may occur as early as 10 weeks post-engraftment, requiring euthanasia by 16 weeks, while others indicate this disease may remain subclinical for longer periods of time [31]. The typical reason for mortality in non-tumor-implanted huNSGS mice is anemia, thrombocytopenia, mast cell and eosinophil hyperplasia, macrophage activation syndrome and/or hemophagocytic lymphohistiocytosis, likely due to the presence and activation of human phagocytes and this occurs in ~30% of huNSGS mice. The culprit is thought to be the overexpression of GM-CSF [36,37]. Utilization of immunodeficient mice such as huNSGS or huNOG-EXL mice with deletion of murine MHC can nearly eliminate the GVHD issue [38,39]. Furthermore, if murine β-2 microglobulin or murine MHC is knocked out in either of these models, there is a reduction in GVHD [40,41]. Another consistent issue with humanized mouse PDX models is the poor representation of myeloid-derived cells in peripheral blood and tissues. The advent of the huNOG-EXL mouse has improved the distribution of myeloid cells in humanized mice [32].

This study demonstrates the successful growth of human melanoma tumor xenografts in humanized mice with sufficient human chimerism. However, we highlight some critical issues that can arise when human chimerism is >80%, for cases where the implanted PDX was first established in a non-humanized mouse. Our initial experiments in NSGS mice showed excellent chimerism in the bone marrow (80–99%), but upon implantation with a melanoma PDX (initially established in a non-humanized NSG mouse), we observed rapid decline and mouse wasting. A similar early onset of GVHD occurred when NSG or NOG mice were engrafted with PBMCs from adult humans [42]. In contrast, when we first established the PDX in a humanized mouse, immune editing could occur, and the resulting evolved PDX did not evoke a toxic T cell response.

While several investigators have utilized PDX tumors implanted into humanized mice, the issue of GVHD or other toxicity development over time is common. If the CD34+ cells engrafted into the immune-deficient mouse are autologous to the PDX, then one might expect the developing CD8+ T effector cells will be less likely to reject the tumor as a result of lack of HLA compatibility. Many of the published reports utilizing humanized PDX mouse models to evaluate the response to therapy used a non-HLA matched source of CD34+ cells to humanize the immune system of the mouse evaluated for PDX response to therapy [42]. In this case, an effort is made to at least match the common HLA serotypes, such as HLA-A2. In fact, the Champion Model has been used for many cancer patients to evaluate the efficacy of specific therapeutics for their tumor tissue [43,44,45,46,47]. In the Champion model, the human PDX is first established in a non-humanized NOG mouse expressing truncated IL-2Rγ, then is implanted into a humanized mouse with verified human chimerism of at least 25% and when possible major HLA matching is attempted. When evaluating the response to ICI, there is undoubtedly a risk that the addition of ICI will amplify the rejection of the tumor by the human T cells based upon HLA mismatch and not T cell recognition of specific tumor antigens. Thus, the therapeutic response in humanized mice might drastically differ from the patient’s response, where there is a complete HLA match between the immune cells and the tumor. Moreover, Kanikarla et al., compared the response of human T cells in humanized mice with either autologous or allogeneic engraftment of T cells [48]. They observed that anti-PD-1 responses to the implanted tumor were better with engraftment of autologous then allogeneic CD34+ PBMC cells. It has also been demonstrated that when undifferentiated CD34+ cells from cord blood are used for engraftment the immune attack of huCD3+ T cells is significantly reduced as compared to engraftment of mature PBMCs. This is because as the huCD3+ T cells developing from the cord blood huCD34+ cells, their interaction with the mouse thymus removes most of the huCD3+ T cells that can recognize mouse antigens [49]. In addition, Morton et al., demonstrated that regression of a PDX tumor in response to ICI in humanized mice requires a high level of chimerism and effective humanization [50].

Human CD34+ hematopoietic stem cells are commercially available. However, the tissue source of these cells should be carefully considered. According to Lonza, one of the leading commercial providers of human stem cells, cord blood-derived CD34+ cells tend to be more naive compared to other tissue sources, which is helpful with HLA mismatch issues. However, the number of cells available per donor is limited. One vial containing 1 million stem cells can generate only 10 humanized mice. For experiments that require large sample sizes, bone marrow CD34+ cells might be a better option due to the large number of CD34+ cells available. However, HLA matching becomes more critical with this source.

We show here that when the human PDX is initially grown in a humanized mouse with chimerism >25%, there is much less toxicity, hence, the long-term survival of many of the mice under study. A drawback of this methodology is the increased expense of establishing the PDX in humanized mice, slower growth of the tumor implanted into humanized than in NSG mice, and the risk that the tumor may not grow or will take a very long time to grow in the humanized mouse. Moreover, the quality of the engraftment and chimerism may be variable such that it is important to first screen donor CD34+ cells to verify that most subtypes of CD45+ cells are represented in blood and tissues in the engrafted recipient mouse. Indeed, slow tumor growth would limit the feasibility of using this method as a co-clinical trial to evaluate responsiveness to ICI combined with personalized targeted therapy. With the methodology utilized here, we were able to perform a long-term experiment and follow tumor response for approximately 7 weeks.

Many technical issues still need to be solved before the use of huNSGS-PDX models in co-clinical trials can be truly effective due to the 4-month expected lifetime. That said, it is plausible that if the engrafted mouse is young (~3–4 weeks), chimerism of >25% is verified after ten weeks, the tumor is implanted and grows out quickly (~30–40 days), and the tumor is allowed to undergo treatment over only a 2–4-week time frame, this reduces the issue of GVHD, fatal disease resembling hemophagocytic lymphohistiocytosis [31], wasting, and rejection of tumor due to HLA incompatibility. Still, the most common reason for terminating the treatment protocol on individual mice was a loss of >20% of the body weight. Because of the frequent animal loss, large numbers of mice are required to achieve a predicted *p*-value of <0.05.

Here, the huNOG-EXL mice were 20 weeks post-engraftment when received and it took ~9 weeks for PDX tumors to grow to a size appropriate for enrolling mice in the treatment regime. Mice remained on treatment for 7 weeks, such that the lifetime for each engrafted mouse enrolled in this study was approximately ~9 months. Fortunately, engraftment remained >25% throughout this study. Based upon results from our study, we propose that future preclinical experiments with humanized mice implanted with human PDX tumors that grow fairly rapidly, and the engrafted mice should have lifespan of at least 7–9 months, with a minimum stable chimerism of >25%. We also propose that for interventional treatments with ICI therapies it is important that there be a good representation of CD3+ T cells in the tumor. Evaluation of the ability of isolated human T cells to respond to stimulation would be helpful, though one must keep in mind that mouse blood availability is quite limited during these experiments, and such an assay would need to be done on a microscale. Alternatively, evaluation of the CD69+ CD3+ T cells using flow cytometry might be possible with the 20 μL bleed used for chimerism determination. While we were unable to fully match HLA genotypes of the engrafted CD34+ cells with the tumor implant, we did match HLA-A2 genotypes between tumor and the initial implantation into huNOG-EXL mice.

The potential value of using humanized mice for preclinical studies to estimate the predicted response to ICI therapy in combination with targeted therapies for patients is extraordinarily high, given the cost of ICI therapy and the variability in the response to new therapeutic regimes that include targeted therapy plus ICI. Our data point to the clear need to establish the initial PDX in a humanized mouse so that there can be appropriate selection of a population of tumor cells that have escaped the T cell response, thus greatly reducing an explosion of human T cell response against the human tumor when it is transplanted into humanized mice for evaluation of therapeutic response. We outline some of the issues around using humanized mouse models to predict the response to ICI therapy and show how it is possible to reduce a toxic immune response to the PDX that results in toxicity, weight loss and eventual death. However, a number of drawbacks are evident, including costs, instability of engraftment of huCD34+ cells, GVHD and toxicity, mouse to mouse variability, and variable engraftment of some donor CD34+ cells. The limitations of our study are the relatively small number of huPDX tumors studied and the failure to directly compare direct implantation of biopsy tissue into both huNSGS and huNOG-EXL mice though this would have not been feasible based on the 4-month expected lifetime of the huNSGS mice. Moreover, we show that for the patient tumor with little infiltration of CD8+ T cells at the time of biopsy, the tumor continued to remain ‘immunologically cold’ when implanted into huNOG-EXL mice even though the stroma of the PDX tumor in humanized mice is completely replaced by mouse stroma during the prolonged period for PDX outgrowth in the mice [27,28,29]. While RGS treatment of mice bearing a huPDX established from a tumor with few CD3+ T cells resulted in a small but significant inhibition of tumor growth, ICI alone and ICI combined with RGS did not induce a therapeutic response in the tumor, even though there was a shift in the number of activated T cells in other tissues in the mouse. These data suggest that the properties of a human tumor that restrict T cell infiltration may be carried forward in the implanted PDX, thus providing a model for studying this important property of tumors.

## 5. Conclusions

Our findings suggest that it is essential to carefully manage immune editing by first establishing PDX tumors in the presence of human immune cells before expanding PDX tumors into a larger cohort of humanized mice to evaluate therapy response. Using this methodology there is less toxicity and GVHD and it is possible to perform preclinical studies using immune checkpoint inhibitors combined with targeted therapies to inform clinical trials and patient response to therapy.

## Figures and Tables

**Figure 1 cancers-15-03695-f001:**
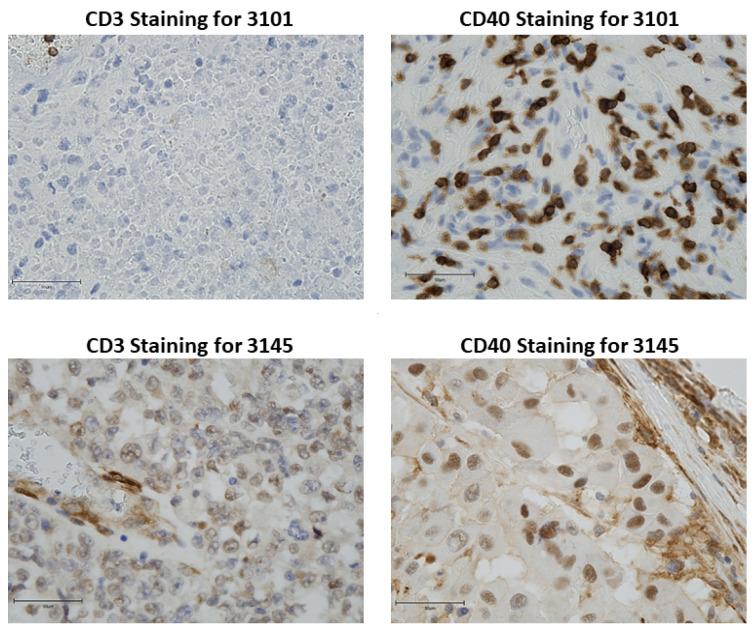
Human melanoma tissue sections were stained for CD3 and CD40. A. Tissue staining for CD3 in FFPE tissue sections 3101 and 3145. B. Tissue staining for CD40 in FFPE tissue sections from patient 3101 and 3145 (magnification 40×) *n* = 2 per staining. Bars represent 50 μm.

**Figure 2 cancers-15-03695-f002:**
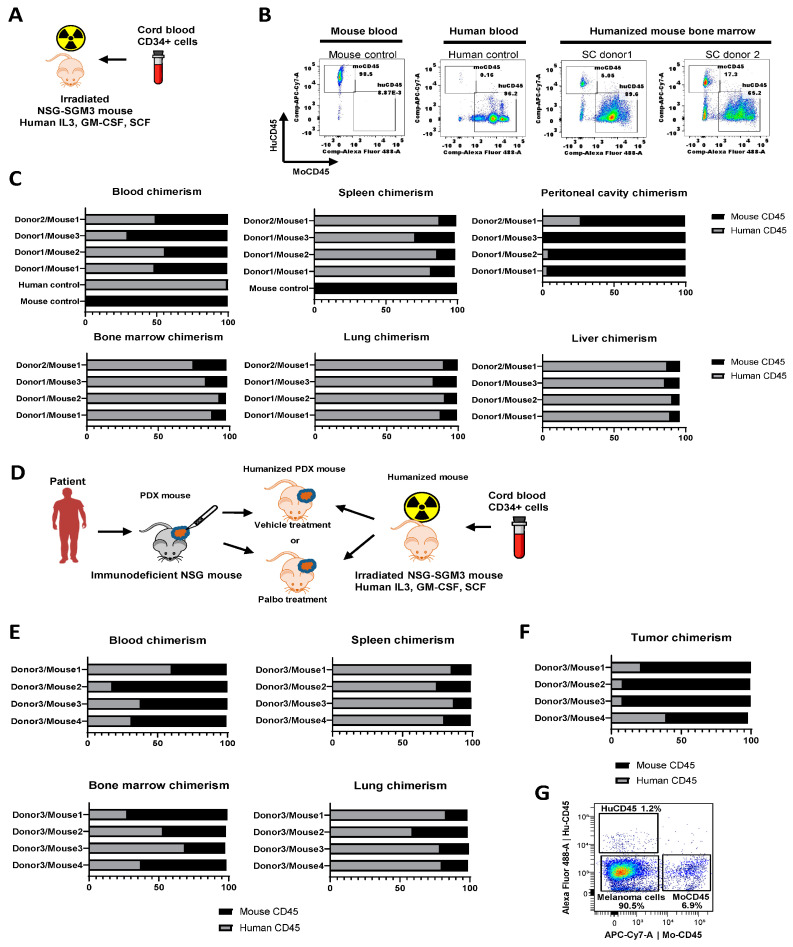
Immune system chimerism in blood and tissues of huNSGS and huNSGS-PDX mice. (**A**) Schematic of the humanization protocol using cord blood stem cells and NSGS mice. (**B**) Representative plots from flow cytometry-based chimerism analysis. Mouse and human blood samples were used as positive controls for human (huCD45) and mouse (moCD45) immune cells, respectively. Stem cell (SC) donor 1 and SC donor 2 indicate bone marrow from mice that were implanted with stem cells from two distinct human donors. Samples were collected at 10 weeks after CD34+ cell engraftment. (**C**) Relative percentages of human and mouse CD45+ immune cells in the blood and indicated tissues of individual humanized mice (*n* = 4 mice). Results from human blood and mouse blood and spleen were included as positive controls. (**D**) Schematic of the humanization, PDX implantation, and drug treatment experiment design. Humanized mice were implanted with human melanoma PDX-1351. (**E**) Relative percentages of human and mouse immune cells in indicated organs obtained from mice described in D (*n* = 4 mice). (**F**) Relative percentages of human and mouse immune cells within the total immune cell infiltrate of the PDX tumors grown in humanized mice shown in D (*n* = 4 mice). (**G**) Representative flow cytometry plots showing percentages of melanoma cells and human and mouse immune cells within the total population of live tumor cells.

**Figure 3 cancers-15-03695-f003:**
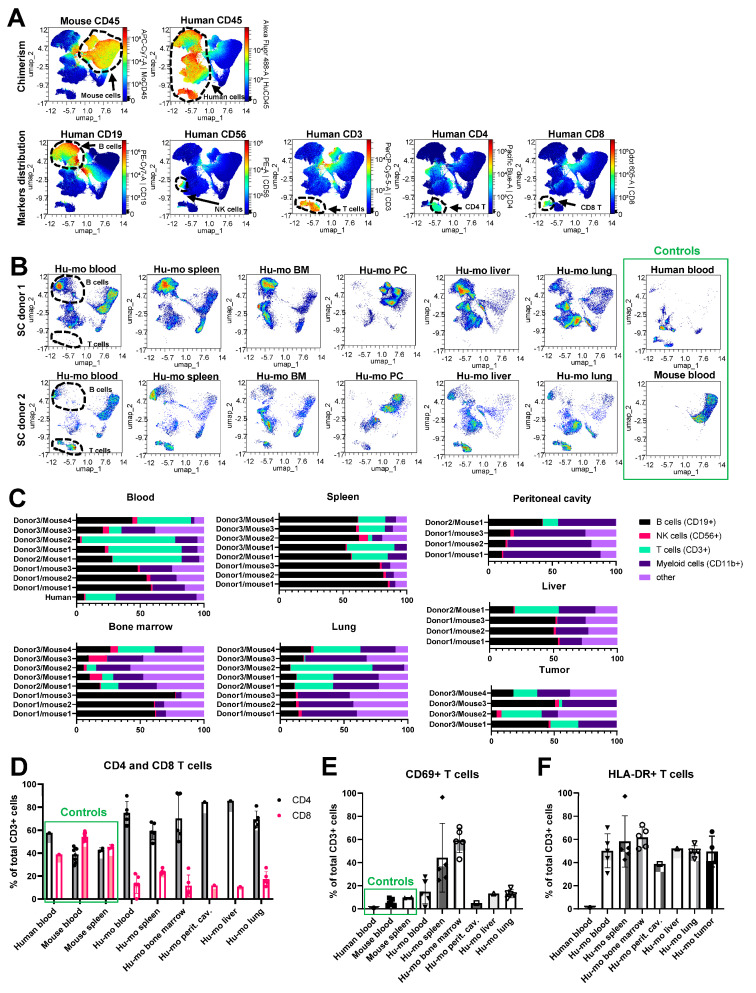
Identification of the human immune cell subsets in humanized mice. (**A**,**B**) Representative UMAP plots of immune cells from humanized mice. Plots were generated in OMIQ based on the surface expression of markers indicated in Appendix A (panel A). (**A**) Heat maps show the levels of expression of indicated lineage markers in all analyzed cells across all samples combined. Key immune subsets identified by marker expression are shown by punctate lines. (**B**) Density plots that indicate the relative abundance of cells across the UMAP in the cell suspensions from indicated organs (**C**) Percentages of CD19+ B cells, CD56+ NK cells, CD3+ T cells, and CD11b+ myeloid cells in indicated tissues of individual humanized mice (*n* = 8 for blood, bone marrow, spleen, and lung, and *n* = 4 for all other tissues). Cells were gated on human CD45+ cells. (**D**) Comparison of CD4+ and CD8+ cell percentages in total population of CD3+ T cells from blood and organs of humanized mice. Human blood and blood and spleen from wild-type C57Bl/6 mice were included for comparison as normal controls. *n* = 5 for blood, spleen, bone marrow, and lung from humanized mice, *n* = 1 for liver and peritoneal cavity lavage in humanized mice and human blood, *n* = 8 for mouse blood, and *n* = 2 for mouse spleen. (**E**) Expression of activation markers CD69 on T cells from indicated tissues of humanized mice implanted with stem cells from donors 2 and 3. Human blood as well as murine blood and spleen from C57Bl/6 mice were included as normal references as in (**D**). (**F**) Same as E, except expression of HLA-DR on T cells was plotted.

**Figure 4 cancers-15-03695-f004:**
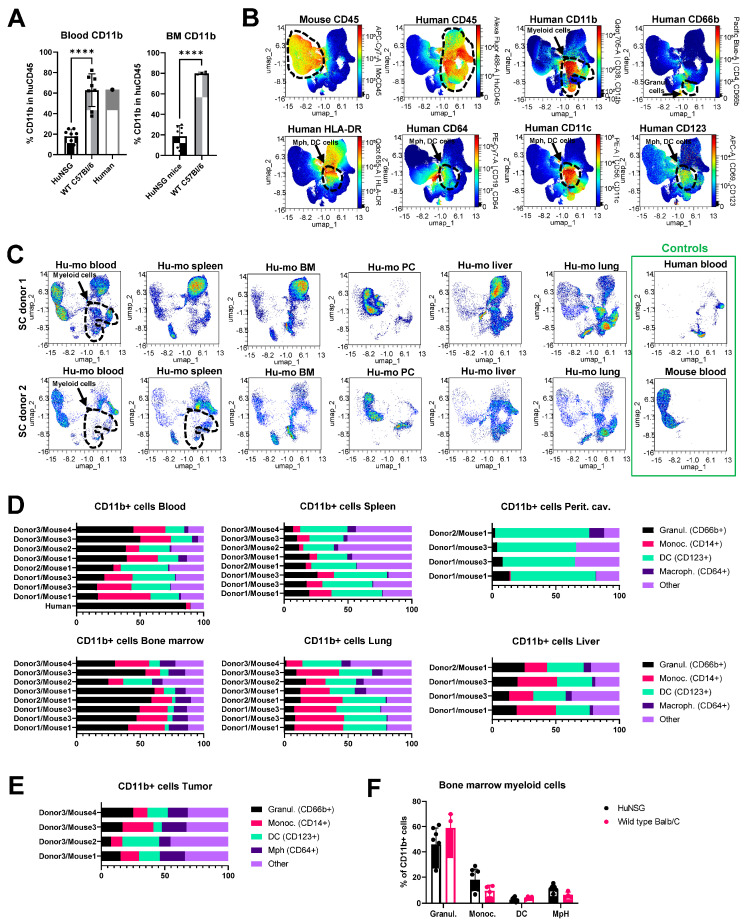
Reconstitution and composition of myeloid cells in humanized mice. (**A**) Comparison of CD11b cell percentages in total leukocytes from blood and bone marrow of humanized mice and wild-type C57Bl/6 mice. *n* = 8 for blood and bone marrow samples from humanized mice, *n* = 8 for mouse blood, *n* = 4 for mouse bone marrow, and *n* = 1 for human blood. Statistical analysis using unpaired *t*-test. **** = *p* < 0.0001. (**B**,**C**) Representative UMAP plots of immune cells from humanized mice. Plots were generated in OMIQ based on the surface expression of markers indicated in Appendix A (panel B). (**B**) UMAP constructed from 33 concatenated samples show the level of expression of indicated markers in all samples combined. Key immune subsets identified by marker expression are shown by punctate lines. (**C**) Density plots showing the relative distribution of cells from indi-cated organs across the UMAP clusters. Three samples were concatenated for donor 1 UMAPs, and one individual sample was plotted for donor 2, and control human and mouse UMAPs. (**D**) Percentages of indicated myeloid cell subsets in blood and tissues of humanized mice. Data are shown as a percent of total human CD45+/CD11b+ cells. *n* = 4 mice for peritoneal cavity and liver graphs, and *n* = 8 for all other tissues. (**E**) Same as D, except tumors from humanized mice were assessed (*n* = 4). (**F**) Comparison of bone marrow composition in huNSGS (*n* = 8) and wild-type Balb/C mice (*n* = 4).

**Figure 5 cancers-15-03695-f005:**
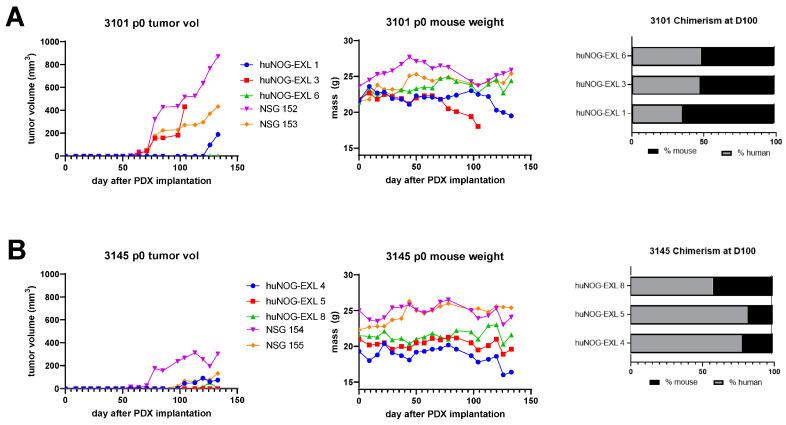
Growth of human melanoma patient-derived xenografts in NSG versus huNOG-EXL mice. (**A**) Left, Human melanoma tumor 3101 volume over time when implanted into humanized (huNOG-EXL) versus immunodeficient (NSG) mice. Middle, an individual mouse mass overtime, showing decline after day 50 in humanized mice. Each line represents an individual tumor (huNOG-EXL mice *n* = 3, NSG mice *n* = 2). Right, chimerism of huNOG-EXL mice bearing PDX 3101 tumors as assessed flow cytometry on human versus murine CD45+ of peripheral blood obtained by a submandibular puncture. All mice demonstrate the persistence of >25% human chimerism (*n* = 3). (**B**) Left, PDX-3145 volume over time when implanted into humanized (huNOG-EXL) versus immunodeficient (NSG) mice. Middle, individual mouse mass overtime, showing decline after day 50 in humanized mice. Each line represents an individual tumor (*n* = 5). Right, chimerism of huNOG-EXL mice bearing PDX-3145 tumors as assessed by flow cytometry on human versus murine CD45+ cells in peripheral blood obtained by a submandibular puncture. All mice demonstrate the persistence of >25% human chimerism (*n* = 3).

**Figure 6 cancers-15-03695-f006:**
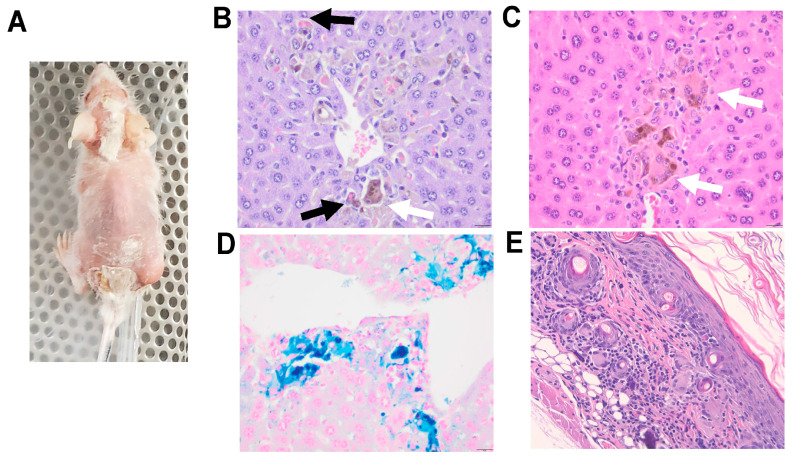
Toxicity observed in huNOG-EXL mice bearing human melanoma tumors. (**A**) Gross alopecia phenotype observed in huNOG-EXL mouse harboring PDX-3101. (**B**,**C**) Macrophage and multinucleated giant cell infiltrates in the liver of 2 animals demonstrating erythrophagocytosis (black arrows) and intracytoplasmic pigment (white arrows); 400× magnification, H&E stain. (**D**) Pearl’s iron stain confirming intracellular pigment is iron hemosiderin, a byproduct of intracytoplasmic breakdown of erythrocytes; 400× magnification, Pearl’s iron. (**E**) Regions of alopecia noted grossly correspond to infiltration and effacement of hair follicles and hair bulbs by lymphocytes, macrophages, and multinucleated giant cells; 400× magnification, H&E stain.

**Figure 7 cancers-15-03695-f007:**
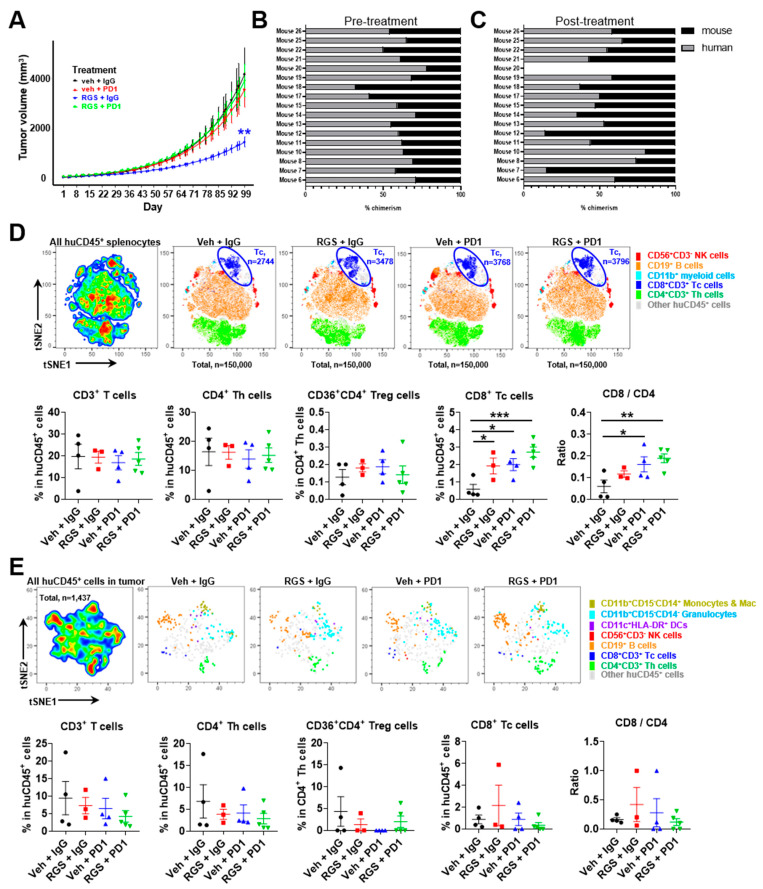
Treatment of huNOG-EXL mice bearing PDX-3101 with RGS and anti-PD-1 antibodies. (**A**) Tumor volume (shown as natural log over time) among mice treated with vehicle + control IgG (black), vehicle + anti-PD-1 (red), RGS + control IgG (blue), or RGS + anti-PD-1 (green). (**B**) Pre-treatment chimerism of huNOG-EXL mice bearing PDX-3101 tumors as assayed by human versus murine CD45+ of peripheral blood by flow cytometry. All mice demonstrate >25% human chimerism (*n* = 17). Blood was collected approximately 17 weeks post-CD34+ cell engraftment. (**C**) Post-treatment chimerism of huNOG-EXL mice bearing PDX-3101 tumors as assayed by human versus murine CD45+ of peripheral blood by flow cytometry. A majority of mice (88%, 14/16) display persistent human chimerism >25% (*n* = 16). Blood was collected approximately 30 weeks post-CD34+ cell engraftment. (**D**,**E**) tSNE analysis of immune profile and percentages of CD3+ T cells, CD4+CD3+ helper T (Th) cells, CD36+CD4+CD3+ regulatory T (Treg) cells, CD8+CD3+ cytotoxic T (Tc) cells, and CD8/CD4 ratio in the spleen (**D**) and tumor (**E**) of individual humanized mice. Cells were gated on human CD45+ cells, n=3-5 mice per treatment group.* = adj *p* < 0.05; ** = adj *p* < 0.01; *** = adj *p* < 0.001.

**Table 1 cancers-15-03695-t001:** Summary of published studies of humanized mice with patient-derived xenografts for evaluation of ICI.

Study	Strain	Ablative Agent	Age at Ablation	Recovery Time	HSC Source and Admin	HSC Number (per Mouse)	First Chimerism Analysis	Mean Chimerism	Tumor Implant	ICI
Kuryk 2018	NOG	chemical	4 weeks	2 days	Cord blood Tail vein injection	60,000	14 weeks	56%	Human melanoma cell lines, SQ	1
Wang 2018	NSG	Irradiation 130cGy	3 weeks	4 h	Fetal liver (Stem Express), unspecified	Not published	12 weeks	>25%	MDA-MB-231s or minced tumor, Intra-fat pad	2
Ashizawa 2017	NOG-dKO	Irradiation 2.5Gy	8 weeks	N/A	Glioma patients Tail vein injection	10^7^	4, 6, 8 weeks	30% (blood) 60% (spleen)	Human lymphoma or glioblastoma cell lines, SQ	3
Tsoneva 2017	NSG	Irradiation 0.8Gy	newborn	3 h	Cord blood Intra-hepatic	180,000–300,000	8 weeks	n/a	A549 cells lung cancer cells	n/a
Taconic	NOG-EXL	Irradiation, 55cGy	2–6 weeks	4–24 h	Cord blood Tail vein injection	40,000	6 weeks	>25% to sell, average >40%	Surgical implant	n/a
Luo 2023	NCG	Irradiation, 200cGy	4–6 weeks	24 h	Cord blood Tail vein injection	100,000	8–20 weeks	>15%	Advanced Gastric Cancer	3

1. Pembrolizumab (Merck) 200 μg or 400 μg day 1, 3, 5 then q3-4d thereafter; 2. Pembrolizumab (Merck) 10 mg/kg x1 then 5 mg/kg q5 days—IV or IP; 3. Anti-PD-1 antibody (In house—a nivolumab biosimilar product) 2 mg/kg.

**Table 2 cancers-15-03695-t002:** Characteristics of melanoma tumors used as PDX implants in humanized mouse studies.

Gene	Melanoma 0287-1351	Melanoma 0287-3101	Melanoma 0287-3125	Melanoma 0287-3145	Protein Alteration
ARID1A	0/1	0/0	0/0	0/0	A226P, P227Q, A349V, A963T, N1827D
BRAF	0/1	0/0	0/0	0/0	S614F
BRAF	0/1	0/0	0/1	0/0	**V600E**
BRCA1	0/1	0/0	0/0	0/0	A532V
CDKN2A	0/0	0/1	0/0	0/0	P63R
KDR	0/1	0/0	0/0	0/0	D994E, P992S
MET	0/0	0/1	0/0	0/0	S308F
MLH1	0/1	0/0	0/0	0/0	S451fs, S46N, I50V, T29A, S30A, E83Q, H285R, N404S
NF1	0/1	0/0	0/0	0/0	D287N
NF1	0/0	0/1	0/0	0/0	Q2239X
NRAS	0/0	0/0	0/0	0/0	wt
PARD3	0/1	0/0	0/0	0/0	S1214F, A499V, E276Q
PARD3B	0/0	0/1	0/0	0/0	D287N
POT1	0/1	0/0	0/0	0/0	P569S
PPP6C	0/1	0/0	0/0	0/0	R148Q
PTEN	0/0	0/0	0/0	0/0	wt
TP53	0/0	0/0	0/0	0/0	wt
TP63	0/0	0/1	0/0	0/0	A596V
ZEB1	0/1	0/0	0/0	0/0	G307E
FGFR1OP	0/0	0/1	0/0	0/1	G106D
FGFR2	0/1	0/0	0/0	0/0	A333V
FGFR2	0/1	0/0	0/0	0/0	A333T
FGFR3	0/1	0/0	0/0	0/0	S354R
FGFR3	0/1	0/0	0/0	0/0	P700S
**Age**	62	67	64	44	
**Sex**	F	M	F	M	
**Stage**	IIIB	IV	IIIC	IIIC	
**Tumor location**	Left abdomen	Axillary lymph node	Inguinal lymph node	Axillary lymph node	
0/0	wt				
0/1	Heterozygous				
1/1	Homozygous				

## Data Availability

No additional data are available not reported here.

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
