# Peer review of "Generation of Orthotopic Patient-Derived Xenografts in Humanized Mice for Evaluation of Emerging Targeted Therapies and Immunotherapy Combinations for Melanoma"

_cancers, 2023, doi:10.3390/cancers15143695_

Round 1
Reviewer 1 Report
In the present manuscript entitled "Generation of Orthotopic Patient-Derived Xenograft in Humanized Mice for Evaluation of Emerging Targeted Therapies and Immunotherapy Combinations for Melanoma", the establishment of two ways of performing PDX in humanized mice is compared. The hypothesis and approach of the study is adequate as well as literature review. However, there are deficiencies in the way the results are presented.
Introduction:
- It should be well defined what GVHD is and the problem it entails when performing PDX's.
- The aim of the study should be clarified without providing results of the study.
- Table 1 is blurred
Material and Methods
- The tumor samples are very few (n=3) so the conclusions the authors provide cannot be conclusive. Moreover, the samples used are very heterogeneous and some of them come from previously treated patients. Doesn't this make the engrafment difficult?
- The medical history of the patients should be summarized as it is not relevant to the study. Do you have any informed consent to give all this information?
- The total number of animals used should be entered, as well as the number of animals used in each experiment.
- How are the mice housed? dimensions of the box? Do you use any type of environmental enrichment? Information on the housing of the mice should be included to comply with transparency in animal experimentation.
- How long were the mice treated? what was the duration of the treatment experiment?
-Table 2 should be included in the patient samples section instead of mouse studies.
-Line 227--> Should introduce "human CD3" instead of "CD3+ human T cells".
- In the immunohistochemistry section, the evaluation criteria for the different immunostains should be summarized.
- Line 243 --> delete Figure 7A
Results:
The results are very long and redundant, and part of them are not contemplated in discussion so they should be summarized.
The authors should limit to present the results and not support them with references since this is already done in the discussion section.
Author Response
Reviewer 1
Reviewer 1 wrote that “The hypothesis and approach of the study is adequate as well as literature review”.
There were several specific comments brought up:
1. Introduction:
- It should be well defined what GVHD is and the problem it entails when performing PDX's.
Response: In the introduction of the revised manuscript we defined GVHD as follows: GVHD occurs when the donor T cells attack recipient tissues and this is followed by antigen presenting cell activation, expansion of alloreactive T cells, and a cytokine storm that results in tissue injury and destruction[13] . In the methods of the revised manuscript we described the criteria and method for assessing GVHD. This is on page 6 of the revised manuscript.
1. The aim of the study should be clarified without providing results of the study.
Response: We would like to thank the reviewer for pointing this out. We have included these points in the Introduction on page 3 of the revised manuscript to read as follows: “The aim of the present study is to develop and characterize a PDX model in humanized mouse, with manageable toxicity, long-term human melanoma tumor growth, maintenance of human chimerism, that is suitable for testing response to immune based therapies.”
- - Table 1 is blurred
Response: We have replaced Table 1 with a high-resolution format - 2. Material and Methods
- The tumor samples are very few (n=3) so the conclusions the authors provide cannot be conclusive. Moreover, the samples used are very heterogeneous and some of them come from previously treated patients. Doesn't this make the engraftment difficult?
Response: We agree that the small sample size is one of the limitations of our study and we mention this in the discussion. Studying tumor responses using humanized PDX models is most useful for patients that have
been previously treated and progressed on treatment. The prior treatments do not usually impair engraftment as we normally get 75% engraftment into NSG mice and NSG-SGM3 mice. - - The medical history of the patients should be summarized as it is not relevant to the study
Response: We have revised the description of patient history to provide only a summary that does not include
any PI. All patients gave informed consent prior to surgery.
- The total number of animals used should be entered, as well as the number of animals used in each experiment.
Response: We have included the n (number of animals) for each experiment in the revised manuscript.
- How are the mice housed? dimensions of the box? Do you use any type of environmental enrichment? Information on the housing of the mice should be included to comply with transparency in animal experimentation.
Response: We use standard HEPA-filtered sterile caging (interior cage height of 5.5 inches and a 75 square inch interior floor area) that is recommended for immune compromised mice with sterile water and sterile food. Mice are given diet gel treats if they show loss of weight. We also include cardboard huts for enrichment. We include this information in the revised manuscript in the Materials and Methods.
- How long were the mice treated? what was the duration of the treatment experiment?
Response: In our optimized huNOG-EXL-PDX mice model, 17/20 (85%) mice demonstrated sufficient tumor growth and health to undergo therapeutic treatment. “We were able to perform a long-term experiment where
we analyzed therapeutic responses in these humanized mice implanted with human melanoma tumors for an average of 49 days.” This information is included in the Discussion on page 19 of the revised manuscript.
-Table 2 should be included in the patient samples section instead of mouse studies.
Response: Thank you, this has been corrected.
-Line 227--> Should introduce "human CD3" instead of "CD3+ human T cells".
Response: Thank you this has been corrected.- - In the immunohistochemistry section, the evaluation criteria for the different immunostains should be summarized.
Response: H&E and immunohistochemical stains were reviewed by a board-certified pathologist (VW). We have included the evaluation criteria in the Method on page 6 of the revised manuscript to read as follows:
“Evaluation criteria for melanoma metastases included cytologic and architectural atypia consistent with melanoma and positive staining for the combined anti-human Melan A/Mart1 rabbit monoclonal antibody red
stain.”
- Line 243 --> delete Figure 7A
Response: Thank you. This has been corrected.
3. Results:
The results are very long and redundant, and part of them are not contemplated in discussion so they should be summarized. The authors should limit to present the results and not support them with references since this is already done in the discussion section.
Response:
We have removed discussion comments and references from the results section when appropriate.

Reviewer 2 Report
The authors present their findings on different CD34+ stem cell transplanted humanized mice in the context of tumor transplantations. The final data is a comparison between established anti-PD1 immuntherapy with the addition of rigosertib.
The main message of the paper appears to be that it is important to check whether a tumor is tolerated by the human immune system in the stem cell transplanted mice, before proceeding to expanding the tumor to larger cohorts of NSG mice.
In the simple summary they claim to have optimized a protocol for evaluating the response to therapy. Hwoever, there are no comparisons of different experimental settings and the results appear to be mostly anecdotal.
Major critiques
1. Please argue how Figure 1-5 bring any novelty compared to current literature on CD34+ transplanted PDX mice and general best practices. These figures mainly present validating data of standard transplantations without added novelty.
2. The discussion includes several important observations but it very long for the limited robustness of the data, and is mostly speculative
3. In the final experiment with testing the tumor in SC transplanted mice, they end up purchasing mice from Taconic? Why did they not use their own that they just spend all this time optimizing on?
4. following point 3: They cannot exclude that their transplantation procedure is just not as good as the commercially available purchased mice or the donor is more tolerant, since they do not compare the same human donors. In the manuscript, they correctly state that both the humanization robustness, the cell types as well as the immune interaction with tumor is highly variable between donors and tumors. Thus the study is severly underpowered to conclude any major improvements of model 1 vs model 2.
Minor critiques
1. Table 1 Summary is not fully exhaustive. Also the quality of the table is extremely low resolution.
2. Poor language in places. Eg. line 516-517: These data indicate that, in contrast to experiments with PDX tumors that were initially established in a non-humanized NSG mouse prior to implantation into humanized NSGS mice, where nearly 100% of mice exhibited wasting.
3. Figure 7A – what do they mean with “predicted mean and SE”? Is this really log fold? The growth curves look surpringsly smooths for a pdx model.
4. Appendix A appears to be only template text
5. No live dead stain human panel B? for myeloid cells Supplementary table 1
6. Why do they switch between UMAP and tSNE plots for the same type of data?
Please proofread
Author Response
Major critiques
- Please argue how Figure 1-5 bring any novelty compared to current literature on CD34+ transplanted PDX mice and general best practices. These figures mainly present validating data of standard transplantations without added novelty.
Response: Figure 1 is important because we hypothesize that response to anti-PD1 and rigosertib may be associated with the presence of CD3+ T cells in the tumor microenvironment and rigosertib has been shown to induce expression of CD40 on tumor cells and infiltration of CD3+ T cells into the TME as part of the mechanism for response to rigosertib-mediated tumor cell death. We are not aware that prior humanized PDX studies have characterized the relationship between CD3 and CD40 expression in tumor tissue and response to therapy. Yes, others have shown CD3 staining, but not CD40 staining of tumor cells as a potential property for response to therapy.
We think that Figure 2 is important to show because it verifies the degree of long term chimerism implant in blood, spleen, bone marrow, lung, and tumor that we are able to achieve with and without the PDX implantation in our humanized mice. We show variation from mouse to mouse and from CD34 cell donor to donor. These data are important to establish when analyzing why GVHD is occurring when we initially establish the PDX in a non-humanized mouse versus a humanized mouse.
Figure 3 details the full phenotype of the human immune cells in the mice and clearly establishes that we get engraftment of both myeloid cells and lymphoid cells. Often it is challenging to get myeloid cell engraftment in humanized mice. However, we do see that there is variability from donor to donor, with donor one CD34+ cells not being as capable of generating T cells. Again, these data are an important part of the full characterization of the immune phenotype of the NSG-SGM3 mice with PDX implants in this study.
Figure 4 is important to show as it demonstrates that both donor 1 and donor 2 are fully capable of generating multiple types of myeloid cells that are present in the blood, spleen peritoneal cavity, bone marrow, lung, liver, and tumor. While in many humanized mouse models the myeloid population is quite deficient, we are achieving reasonable representation of granulocytes, monocytes, DC, macrophages though there was some variability from donor to donor and mouse to mouse. These are important data to determine which donor to choose for the PDX experiments and to verify the humanization. level in the model.
Figure 5 demonstrates that when compared to outgrowth in NGS mice, PDX tumor growth in humanized NOG-EXL mice implanted with tumor from patients 3101 and 3145 was considerably reduced, with the exception of mouse three in the PDX 3101 humanized NOG-EXL model which grew equivalently to the PDX tumors implanted in NSG mice. There was a trend toward more weight loss in the mice implanted with PDX tumors in the NOG-EXL humanized mice as compared to NSG mice—except for HuNOG-EXL mouse 6 with PDX from patient 3101 which failed to grow in the humanized model and consequently did not exhibit weight loss. This was in contrast to the humanized NOG-EXL mice where tumors grew and there was loss of weight by 80 or 100 days post implantation (mice 3 and 1, respectively). These mice maintained chimerism of > 40% out to 100 days.
- The discussion includes several important observations but it very long for the limited robustness of the data and is mostly speculative.
Response: We have revised the discussion extensively and removed speculative points.
- In the final experiment with testing the tumor in SC transplanted mice, they end up purchasing mice from Taconic? Why did they not use their own that they just spend all this time optimizing on?
Response: We would like to thank the reviewer for this question! In the final humanized PDX mouse experiment testing drug response, to perform the experiments in NSG-SGM3 mice that we humanized in our lab, this would have required a large batch of CD34+ cells from the same patient sufficient to produce more than 30 humanized mice. The mice would need to be subsequently verified for chimerism, likely resulting in elimination of a number of mice to finally select 20 sufficiently humanized mice with good engraftment. Moreover, according to our hypothesis, the PDX tumors needed to be established first in humanized mice. To ensure quality control, we decided that using humanized mice from Taconic where chimerism was verified and consistent with regard to donors was the most efficient way to move forward. Therefore we ordered humanized mice from Taconic to first evaluate CD34+ cells from different donors as to the ability of PDX tumors to grow in these humanized mice. Based upon the results from these experiments we were able to identify which PDX tumors would grow in humanized NOG-EXL mice, and then order 20 humanized NOG-EXL mice of the same age with the same CD34+ cell donor to implant PDX tumors that had been established in humanized mice, thus significantly reducing any immune shock upon PDX implantation—since the PDX had been established in the presence of a human immune system and any cells that were immune-reactive would have been edited out. While we did not perform a head-to-head comparison between the humanized mice developed using in-house methodology and the Taconic mice, the point of our present work is that best results are obtained when one carefully manages immune editing by first establishing PDX tumors in the presence of human immune cells before expanding PDX tumors into a larger cohort of humanized mice to evaluate therapy response. With this approach there is reduced GVHD and increased lifespan of the mice for therapeutic studies.
- following point 3: They cannot exclude that their transplantation procedure is just not as good as the commercially available purchased mice or the donor is more tolerant, since they do not compare the same human donors. In the manuscript, they correctly state that both the humanization robustness, the cell types as well as the immune interaction with tumor is highly variable between donors and tumors. Thus the study is severely underpowered to conclude any major improvements of model 1 vs model 2.
Response: Thank you for this comment. The point of our paper is not that our transplantation procedure is not as good as commercially available purchased mice. The point is that first establishing the PDX in a humanized mouse is preferable to establishing in non-humanized mice then engrafting to humanized mice. We also show the importance of comparing a number of donors for engraftment prior to starting preclinical trials to ensure the selection of the best donor for CD34+ cell engraftment.” These issues complicate the potential utility of humanized mouse studies for preclinical trials.
We agree with the reviewer that our present study has limitations. These points were included in the Discussion on page 20 of the revised manuscript to read as follows: “The limitations of our study are the relatively small number of huPDX tumors studied and the failure to directly compare direct implantation of the same biopsy tissue into both huNSG-SGM3 and huNOG-EXL mice.”
Minor critiques
- Table 1 Summary is not fully exhaustive. Also the quality of the table is extremely low resolution.
Response: We have revised and replaced Table 1 with a high-resolution format.
- Poor language in places. Eg. line 516-517: These data indicate that, in contrast to experiments with PDX tumors that were initially established in a non-humanized NSG mouse prior to implantation into humanized NSGS mice, where nearly 100% of mice exhibited wasting.
Response: This sentence has been corrected to read as follows:
“ These data indicate that, in contrast to experiments with PDX tumors that were initially established in a non-humanized NSG mouse prior to implantation into humanized NSGS mice where nearly 100% of mice exhibited wasting. only 15% of the mice with PDX implants that were established in huNOG-EXL mice exhibited sufficient wasting to limit their enrollment in this treatment study.”
- Figure 7A – what do they mean with “predicted mean and SE”? Is this really log fold? The growth curves look surprisingly smooths for a PDX model.
Response: These are model based “predicted values” transformed to the original scale of the data. Model based estimates smooth over the data and are better than raw data which are more affected by random variation. Though data was analyzed on the log scale to make modelling more straightforward (linear over time and other analysis assumptions), we bring back the curvature when the data are back-transformed to the original scale. We have now used this approach for years and is widely acceptable in both basic and clinical research.
- Appendix A appears to be only template text
Response: We would like to thank the reviewer for pointing this out. The template text of Appendix A is deleted in the revised manuscript.
- No live dead stain human panel B? for myeloid cells Supplementary table 1
Response: Live/Dead Aqua was used as a live dead stain in human panel B.
- Why do they switch between UMAP and tSNE plots for the same type of data?
Response: These experiments were performed by different personnel in the lab. t-SNE and UMAP have the same principle and workflow: create a high dimensional graph, then one is able to reconstruct it in a lower dimensional space while retaining the structure. UMAP uses the Stochastic Gradient Descent (SGD) instead of the Gradient Descent (GD) like t-SNE, which speeds up the computations and consumes less memory and allows the analysis of many more events in a shorter amount of time.

Reviewer 3 Report
The manuscript entitled "Generation of Orthotopic Patient-Derived Xenograft in Humanized Mice for Evaluation of Emerging Targeted Therapies and Immunotherapy Combinations for Melanoma" by Chi Yan et al., was presented as an original article in which the authors reported the development of the PDX model of melanoma and suggested it as a novel tool for personalized medicine. They present a well-written, thoroughly documented description of their experience generating orthotopic PDX models from patients with melanoma. The reviewer fully agrees with the introduction, assessment of the limitations of current in vitro and in vivo models, and approach to establishing the xenografts. Unfortunately, the manuscript falls short of fully characterizing these models, limiting their utility in its current form. In fact, the idea behind the manuscript is great and can be helpful if issues related to the model can be fixed.
Major comments for the figures:
The size of the figures is too small. Immunostaining and cellular characteristics are not evident in the current format. Please, resize and provide a zoom of a representative field. In the original figures, scale bars are missing; add them and specify the magnification (x) and the length (μm) in the Figure Legend. Characters in some boxes are not legible in real size.
Please provide a Supplementary Figure in your Supplementary Information file to graphically account for the FACS sequential gating/sorting strategies or provide gating/sorting strategies in-figure.
Please insert the tables in the document as a modifiable format and not as a figure. The actual quality of the file is very bad.
Author Response
Reviewer 3
Reviewer 3 noted that “the authors reported the development of the PDX model of melanoma and suggested it as a novel tool for personalized medicine. They present a well-written, thoroughly documented description of their experience generating orthotopic PDX models from patients with melanoma. The reviewer fully agrees with the introduction, assessment of the limitations of current in vitro and in vivo models, and approach to establishing the xenografts.”
Major comments for the figures:
The size of the figures is too small. Immunostaining and cellular characteristics are not evident in the current format. Please, resize and provide a zoom of a representative field. In the original figures, scale bars are missing; add them and specify the magnification (x) and the length (μm) in the Figure Legend. Characters in some boxes are not legible in real size.
Response: We replaced figure 1 with 40X magnification for each of the examples and added the magnification and a bar that represents 50μm for each figure.
Please provide a Supplementary Figure in your Supplementary Information file to graphically account for the FACS sequential gating/sorting strategies or provide gating/sorting strategies in-figure.
Response: We have included FACS sequential gating/sorting strategies as Supplementary Figure 1 in the revised manuscript.
Please insert the tables in the document as a modifiable format and not as a figure. The actual quality of the file is very bad.
Response: We have revised and replaced Table 1 with a high-resolution format. Additional modifiable word format of Tables are attached and submitted along with the revised manuscript.

Round 2
Reviewer 1 Report
The authors have taken into account most of the suggestions requested.
How long were the mice treated? what was the duration of the treatment experiment?
Response: In our optimized huNOG-EXL-PDX mice model, 17/20 (85%) mice demonstrated sufficient tumor growth and health to undergo therapeutic treatment. “We were able to perform a long-term experiment where
we analyzed therapeutic responses in these humanized mice implanted with human melanoma tumors for an average of 49 days.” This information is included in the Discussion on page 19 of the revised manuscript.
Response: This information should be included in Material and Methods not in the discussion section.
Author Response
Thank you very much for this suggestion. In the revised version of the manuscript, we have moved the sentence pointed out to Materials and Methods.
Reviewer 3 Report
Most of reviewer's comments were taken into account during the preparation of the revised version of the manuscript. She found the revised version of the manuscript much improved in terms of data presentation and text organization. So, she is convinced by the explanation the authors are presenting and approves the publication in the current state.
Author Response
Thank you for your approval of our manuscript for publication in its current state.